# EXPLAINING MULTICLASS CLASSIFIERS WITH CATEGORICAL VALUES: A CASE STUDY IN RADIOGRAPHY

**Luca Franceschi, Cemre Zor, Muhammad Bilal Zafar, Gianluca Detommaso, Cedric Archambeau, Tamas Madl, Michele Donini & Matthias Seeger**
Amazon Web Services
Berlin, Germany & London, UK
`franuluc@amazon.de` & `cemrezor@amazon.co.uk`

## ABSTRACT

Explainability of machine learning methods is of fundamental importance in healthcare to calibrate trust. A large branch of explainable machine learning uses tools linked to the Shapley value, which have nonetheless been found difficult to interpret and potentially misleading. Taking multiclass classification as a reference task, we argue that a critical issue in these methods is that they disregard the structure of the model outputs. We develop the Categorical Shapley value as a theoretically-grounded method to explain the output of multiclass classifiers, in terms of transition (or flipping) probabilities across classes. We demonstrate on a case study composed of three example scenarios for pneumonia detection and subtyping using X-ray images.

## 1 INTRODUCTION

Machine learning (ML) has emerged as a powerful tool in healthcare with the potential to revolutionize the way we diagnose, treat and prevent diseases. ML algorithms have a wide range of applications including early detection of diseases, risk prediction in patients developing certain conditions, optimisation of treatment plans, improved prognosis, assistance in clinical decision-making, gene expression analysis, genomic classification, improved personalize patient care and more. However, the adoption of ML in clinical practice has often been hampered by the opaqueness of the ML models. This opaqueness may trigger skepticism in clinicians and other end-users such as patients or care-givers to trust model recommendations without understanding the reasoning behind their predictions, which delays and / or decreases the adoption of state-of-the art technologies and hinders further advances.

Various methods have been proposed in the literature to enhance the explainability of ML models (XAI). Among these, (local) feature attribution methods such as SHAP (Lundberg & Lee, 2017) or variants (e.g. Frye et al., 2020; Chen et al., 2018; Heskes et al., 2020) have gained considerable traction. In fact, Shapley value based explanations are the most popular explainability methods according to a recent study by Bhatt et al. (2020) These methods, supported by a number of axioms (properties) such as nullity, linearity and efficiency, provide insight into the contribution of each feature towards the model decisions. There is, however, a growing scrutiny into the utility of these techniques, which are judged to be unintuitive and potentially misleading (Kumar et al., 2020; Mittelstadt et al., 2019), and do not support contrastive statements (Miller, 2019). While part of these issues may be rooted in misinterpretations of the technical tools involved[1], in this paper we argue that a critical flaw in current approaches is their failure to capture relevant structure of the object one wishes to explain (the explicandum). In contrast, we take the position that attributive explanations should comply with the nature of the explicandum: in particular, if the model output is a random variable (RV), we should represent marginal contributions as RVs as well. Our contribution, which we dub *the Categorical Shapley value*, can fully support statements such as "the probability that the feature $x_i$ causes $x$ to be classified as viral pneumonia rather than bacterial pneumonia is $y$", which we develop, experiment and discuss in this paper within the context of X-ray imaging.

---

[1]For instance, the Shapley value is a descriptive rather than prescriptive tool. This means that, in general, one should not expect that changing the feature with the highest Shapley value should lead to the largest change in the outcome.

## 1.1 THE SHAPLEY VALUE AND ITS APPLICAITON TO EXPLAIN MULTICLASS CLASSIFIERS

For concreteness, we focus here on the multiclass classification ($d$ classes) as one of the most common tasks in ML. Let $f : \mathcal{X} \subseteq \mathbb{R}^n \mapsto \mathcal{Y}$ be a (trained) multiclass classifier and $x \in \mathcal{X}$ an input point. One common strategy to explain the behaviour of the model at $x$ is to attribute an importance score to each input feature through the computation of the Shapley value (SV) (Shapley, 1953a). In order to do so, one must first construct a cooperative game $v$ where players correspond to features, and coalitions correspond to features being used: that is $v(S) = f(x_{|S})$, where $S \in 2^n$. [2] Then, for each $i \in [n]$, the Shapley value is given by

$$\psi_i(v) = \sum_{S \in 2^{[n] \setminus i}} p(S)[v(S \cup i) - v(S)] = \mathbb{E}_{S \sim p(S)}[v(S \cup i) - v(S)]; \tag{1}$$

where $p(S) = \frac{1}{n} \binom{n-1}{|S|}^{-1}$ if $i \notin S$ and 0 otherwise. The quantity $v(S \cup i) - v(S)$ is called the marginal contribution of $i$ to the coalition $S$. See Roth (1988) for an in-depth discussion of the SV and related topics.

Historically, the SV has been developed as an answer to the question: How can we assign a worth (or value) to each player $i$? The SV does so by distributing "fairly" the *grand payoff* $v([n])$ among players, so that (1) if a player never contributes to the payoff, their worth is null, (2) if any two players have indistinguishable marginal contributions, they have the same worth, and (3) if $v$ is a linear combination of two games, say $u$ and $w$, then the worth of $i$ for $v$ is the corresponding linear combination of their worth for $u$ and $w$. The game $v$ could typically represent an economic or political process (e.g. a vote) and, critically, would be modelled as a real-valued set function; i.e. $v : 2^d \mapsto \mathbb{R}$, so that $\psi_i(v) \in \mathbb{R}$.

## 2 CATEGORICAL GAMES AND VALUES

In our case, the grand payoff is the output $f(x)$ that determines the class the model assigns to $x$. Whilst in practice $f$ could be implemented in various ways, several modern ML models (e.g. neural networks) output *distributions* over the classes – e.g. through a softmax layer. Equivalently, one may think of $f(x)$ as an $E$-valued (categorical) random variable. Using the one-hot-encoding convention, we identify $E = \{e_s\}_{s=1}^d$ as the one-hot vectors of the canonical base of $\mathbb{R}^d$. Now, however, it becomes unclear which real number should be assigned to a difference of random variables. Moreover, averaging over coalitions $S$, as done in Eq. (1), may induce a semantic gap in this context. To recover the standard pipeline to compute the SV, one may settle for explaining the logits or the class probabilities as if they were independent scalars. However, this may lead to paradoxical explanations that attribute high importance to a certain feature (say $x_1$) for *all* classes, failing to capture the fact that an increase in the likelihood of a given class must necessarily result in an aggregated decrease of the likelihood of the others. Here we show how to avoid such step which causes loss of structure and rather explain $f(x)$ directly as a random variable.

For a player $i$ and a coalition $S$ not containing $i$, we need to relate $v(S)$ with $v(S \cup i)$ in order to quantify the marginal contribution of $i$ to $S$. This relationship is not just in terms of the marginal distributions of these two variables, but also of their dependence. In this paper, we assume a simple dependency structure between all variables $v(S)$, in that $v(S) = \tilde{v}(S, \varepsilon)$ for $\varepsilon \sim p(\varepsilon)$ where $\tilde{v}$ is a deterministic mapping to $E$, and $\varepsilon$ is a random variable distributed according to some $p(\varepsilon)$. Let $v(S)$ be a $d$-way categorical distribution with natural parameters $\theta_{S,j}$, in that

$$\mathbb{P}\left(v(S) = j\right) = \frac{e^{\theta_{S,j}}}{\sum_k e^{\theta_{S,k}}} = \mathrm{Softmax}(\theta_S).$$

We call such $v$ a *Categorical game*. We can implement the aforementioned dependency assumption by the Gumbel-argmax reparameterization (Papandreou & Yuille, 2011): $\tilde{v}(S, \varepsilon) = \arg\max_k\{\theta_{S,k} + \varepsilon_k\}$, where $\varepsilon_1, \ldots, \varepsilon_d$ are independent standard Gumbel variables.

Given this construction, we redefine the *marginal contribution* of $i$ to $S$ as the random variable $\tilde{v}(S \cup i, \varepsilon) - \tilde{v}(S, \varepsilon)$ for $\varepsilon \sim p(\varepsilon)$. This RV takes values in the set $E - E = \{e - e' \,|\, e, e' \in E\}$; we

---

[2] In practice, out-of-coalition features must often be given a value; this could be an arbitrary baseline, a global or a conditional average Sundararajan & Najmi (2020); Aas et al. (2021).

shall call its distribution

$$q_{i,S}(z) = \mathbb{P}(v(S \cup i) - v(S) = z \mid S), \quad z \in E - E.$$

Note that $q_{i,S}(x)$ is a conditional distribution, given $S \in 2^{[n] \setminus i}$ and $E - E$ is a set containing $0 \in \mathbb{R}^d$ and all vectors that have exactly two non-zero entries, one with value $+1$ and the other $-1$.

We can view this construction as a generalized difference operation $v(S \cup i) \ominus v(S)$ between random variables rather then deterministic values, where the $\ominus$ incorporates the above dependency assumption. We define our *Categorical Shapely value* as the random variable $\xi(v) = \{\xi_i\}_{i \in [n]}$, where

$$\xi_i(v) = v(S_i \cup i) \ominus v(S_i) = \tilde{v}(S \cup i, \varepsilon) - \tilde{v}(S, \varepsilon) \qquad \text{for } \varepsilon \sim p(\varepsilon) \text{ and } S \sim p(S). \quad (2)$$

Note these RVs have multiple sources of randomness, which are independent from each other. We can marginalise over $p(S)$ to obtain the distribution $q_i(x)$ of $\xi_i(v)$: for every $z \in E - E$:

$$q_i(z) = \mathbb{P}(\xi_i(v) = z) = \mathbb{E}_{S_i \sim p^i}[q_{S_i,i}(z)] = \sum_{S_i \in 2^{[n] \setminus i}} p(S_i) q_{S_i,i}(z). \quad (3)$$

One major advantage of this novel construction is that now the distribution of the Categorical SV is straightforward to interpret. Indeed, the probability masses at each point $z = e_r - e_s \in E - E$ are interpretable as the probability (averaged over coalitions) that player $i$ causes the payoff of $v$ (and hence the prediction of $f$ to flip from class $s$ to class $r$. We refer to $q_i(e_r - e_s)$ as the *transition probability* induced by feature $i$.

Interestingly, we can derive a closed form analytical expression for the $q_{i,S}$ and, hence, for the $q_i$. We do this in Section A. The following proposition relates the Categorical Shapley value with the standard SV and gives a number of properties that can be derived for the categorical SV.

**Proposition 2.1.** *Let $\xi$ be the Categorical Shapley value defined in equation 2. Then:*

1. $\mathbb{E}[\xi_i(v)] = \psi_i(\mathbb{E}[v]) \in [-1, 1]^d$, *where $\mathbb{E}[v]$ is the $n$-players game defined as $\mathbb{E}[v](S) = \mathbb{E}[v(S)] = \text{Softmax}(\theta_S)$;*

2. *If $i$ is a null player, i.e. $v(S \cup i) = v(S)$ for all $S \neq \emptyset$, then $\xi_i(v) = \delta_0$, where $\delta_0$ is the Dirac delta centered in $0 \in \mathbb{R}^d$;*

3. *If $v = v'$ with probability $\pi \in [0, 1]$ and $v = v''$ with probability $1 - \pi$ (independent from $S$), then $q_i(z) = \mathbb{P}(\xi_i(v) = z) = \pi \mathbb{P}(\xi_i(v') = z) + (1 - \pi)\mathbb{P}(\xi_i(v'') = z) = \pi q_i'(z) + (1 - \pi)q_i''(z)$.*

4. $v([n]) \ominus v(\emptyset) = \sum_{i \in [n]} \mathbb{E}_{S \sim p(S)}[\xi_i(v)]$, *where the sum on the right hand side is the sum of (dependent) $E - E$-valued random variables.*

Property 1 essentially shows that the Categorical SV is strictly more expressive than the traditional Shapley values, whilst putting the traditional SVs for multiclass classification under a new light. Properties 2, 3, and 4 may be seen as the "adaptations" to the Categorical SV of the null player, linearity and efficiency axioms, respectively. In particular, the standard linearity axiom would be of little consequence in this context as taking a linear combination of categorical RVs does not lead to another categorical RV. Instead, Property 3 addresses the common situation where the classifier one wishes to explain is a (probabilistic) ensemble, relating the distributions of the respective Categorical SVs. See Section C for a brief discussion of the related work in the cooperative game theory literature.

## 3 Detecting Pneuomonia in Chest X-Rays: A Case Study

This section employs the Categorical SV (CSV) to analyse a commonly used deep learning architecture, ResNet-18 (He et al., 2015) for pneumonia detection and subtyping using X-ray images, which is casted as a multiclass classification problem based categorising subjects into three classes: healthy controls (HC - class 0), bacterial pneumonia cases (BP - class 1) and viral pneumonia cases (VP - class 2). The model has been trained on chest X-ray images collected from pediatric patients, aged one to five, as part of their routine clinical care in Guangzhou Women and Children's Medical Center (Kermany et al., 2018). The aim is to show the importance of using structured explanations

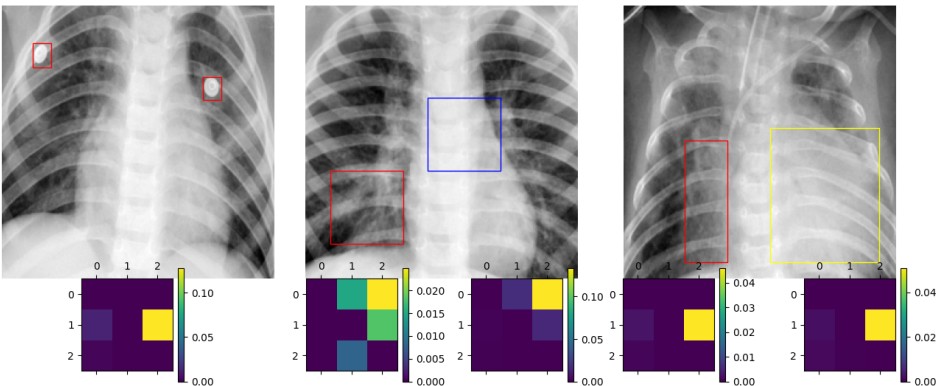

Figure 1: Three example subject X-ray images and Categorical Shapley values relative to the depicted patches, plotted as matrices. (Left) Ground-truth: VP Prediction: BP. Patch representing two artifacts which should not impact the model decision (Center) Ground-truth: VP Prediction: VP. Two patches, the red one on the left highlighting a section where pneumonia is visible and blue depicting a patch in the middle mediastinum. (Right) Ground-truth: VP Prediction: BP. The red patch is related to a pneumonia area, the yellow one highlights the heart region the patient.

even when the model is fine-tuned to the problem of interest, in this case with a mean balanced accuracy score of $84.7\%$. We select three example scenarios (as depicted in Figure 1) to analyse different use-cases where CSV empowers the decision process.

**Case One: Artifacts** Figure 1 (Left) shows an example scenario of an image with artifacts as depicted in red bounding boxes. The probability distribution output by the model for the ground-truth class BP and the predicted class VP are given as $47.8\%$ and $48.0\%$ respectively. CSV measures the transition probability from VP to BP, generated by the artifact regions, as $12.7\%$, which implies the presence of these artifacts as a root cause behind the confusion between BP and VP.

**Case Two: Correct Classification** Figure 1 (Center) shows a correctly classified VP. However, even though the main affected area in this patient is depicted by red by independent experts, the contribution of this area to the decision has been found negligible (around $1\%$, see the left matrix under the Center image), making the model's recommendation untrustworthy. Furthermore, the transition probability from VP to HC calculated for the middle mediastinum region (depicted in blue), which is not expected to be a region of interest for pneumonia, can found as high as $13.3\%$, flagging this region as incorrectly important for the decision process of the model.

**Case Three: Incorrect Classification** When the incorrectly classified case shown in Figure 1 (Right) is analysed, the transition probability for the area in red, which is labelled as a main affected area of VP by independent experts, from the prediction class BP to the ground-truth class VP is calculated as zero. On the other hand, the heart region identified by yellow is shown to exhibit over $5\%$ transition probability to the VP and BP classes, although this value would be expected to be close to zero. Both of these findings help highlight inconsistencies in the behaviour of the model.

## 4 DISCUSSION AND CONCLUSION

By analysing three example scenarios in Section 3, we have underlined the importance of using model explainability even for fine-tuned, seemingly highly performing models, especially for use in critically important application areas such as healthcare. Employing categorical games and values empowers a structural understanding of the multiclass classification problem by providing information about transition probabilities across classes informing about "decision flips", in addition to the feature contribution information obtained from classical methods. Such knowledge would highly strengthen the model design process; e.g. by promoting the use of comprehensive pre-processing steps, ensemble classification designs or intelligent model tuning.

While we implement a case study on pneumonia classification using X-ray images as a proof-of-concept, the method proposed is extendable to all modalities including genomics, free-text or tabular data. For out of coalition portions of the image, we employed a simple background constant value. We plan to consider more sophisticated formulations in the future. Another invaluable path for future work includes developing better visualization and summarization methods as well as interactive interfaces to support clinicians and other end-users.

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

## A  ANALYTIC EXPRESSION OF THE PDF OF CATEGORICAL DIFFERENCES

Consider $E = \{e_1, \ldots, e_d\}$ with $d \geq 3$. Suppose that $v(S)$ has a $d$-way categorical distribution with natural parameters $\theta_{S,j}$, in that

$$\mathbb{P}\left(v(S) = j\right) = \frac{e^{\theta_{S,j}}}{\sum_k e^{\theta_{S,k}}}.$$

Categorical games emerge, e.g., when explaining the output of multiclass classifiers or attention masks of transformer models (Kim et al., 2017; Vaswani et al., 2017).

A latent variable representation is given by the Gumbel-argmax reparameterization (Papandreou & Yuille, 2011):

$$\tilde{v}(S, \varepsilon) = \arg\max_k \{\theta_{S,k} + \varepsilon_k\},$$

where $\varepsilon_1, \ldots, \varepsilon_d$ are independent standard Gumbel variables with probability distribution function $p(\varepsilon_j)$ and cumulative distribution function $F(\varepsilon_j)$ given by

$$F(\varepsilon_j) = \exp\left(-e^{-\varepsilon_j}\right), \quad p(\varepsilon_j) = \exp\left(-\varepsilon_j - e^{-\varepsilon_j}\right).$$

At this point, assume that $e_j = [\mathbf{1}_{k=j}]_k \in \{0,1\}^d$ are the standard basis vectors of $\mathbb{R}^d$. Then, $E - E = \{e_r - e_s \mid 1 \leq r, s \leq d\}$ has size $d^2 - d + 1$, and the distribution of $v(S \cup i) \ominus v(S)$ is given by the off-diagonal entries of the joint distribution $Q_{i,S}(r, s) = \mathbb{P}(v(S \cup i) = r, v(S) = s)$.

We can work out $Q_{i,S}(r, s)$ explicitly. Denote

$$\alpha_j = \theta_{S \cup i, j}, \quad \beta_j = \theta_{S,j}, \quad \rho_j = \alpha_j - \beta_j.$$

Without loss of generality, we assume the categories to be ordered so that $\rho_1 \geq \rho_2 \geq \cdots \geq \rho_d$. Then:

$$\tilde{Q}_{i,S}(r, s) = e^{\alpha_r + \beta_s} \left(C_s - C_r\right) \mathbf{1}_{r<s} \quad (r \neq s),$$
$$\tilde{Q}_{i,S}(r, r) = e^{\beta_r - \bar{\beta}_r} \sigma\left(\bar{\beta}_r - \bar{\alpha}_r + \rho_r\right) \mathbf{1}_{r<d} + e^{\alpha_d - \bar{\alpha}_d} \mathbf{1}_{r=d},$$

where

$$\bar{\alpha}_k = \log \sum_{j=1}^k e^{\alpha_j}, \quad \bar{\beta}_k = \log \sum_{j=k+1}^d e^{\beta_j},$$
$$c_k = e^{-\bar{\beta}_k - \bar{\alpha}_k} \left(\sigma\left(\bar{\beta}_k - \bar{\alpha}_k + \rho_k\right) - \sigma\left(\bar{\beta}_k - \bar{\alpha}_k + \rho_{k+1}\right)\right),$$
$$C_t = \sum_{k=1}^{t-1} c_k, \quad \sigma(x) = \frac{1}{1 + e^{-x}}.$$

The derivation is provided in Appendix B. We write $\tilde{Q}_{i,S}$ instead of $Q_{i,S}$ due to the specific ordering of categories. The induced distribution of $v(S \cup i) \ominus v(S)$ is

$$\sum_{r<s} \tilde{Q}_{i,S}(r,s)\delta_{e_r-e_s} + \left(\sum_r \tilde{Q}_{i,S}(r,r)\right)\delta_{\mathbf{0}},$$

from which the off-diagonal entries of $\tilde{Q}_{i,S}(r,s)$ can be reconstructed.

Assume that $Q_{i,S}(r,s)$ are given for all $S$ in a common ordering of the categories, in that $Q_{i,S}(r,s) = \tilde{Q}_{i,S}(\pi_S(r), \pi_S(s))$, where $\pi_S$ is a permutation of $\{1, \ldots, d\}$ fulfilling the ordering condition used above. If

$$Q_i(r,s) = \mathbb{E}_{S \sim p^i}\left[Q_{i,S}(r,s)\right],$$

the distributions of Categorical values are given by

$$q_i = \sum_{r,s} Q_i(r,s)\delta_{e_r-e_s}.$$

The probability masses at each point $e_r - e_s \in E - E$ are interpretable as the probability (averaged over coalitions) that player $i$ causes the payoff of $v$ to flip from class $s$ to class $r$.

We may define the following *query functional* on top of this distribution is

$$\ell_{\mathrm{mc}} = \max_s \sum_{r \neq s} Q_i(r,s),$$

which quantifies the largest probability of any change in the output led by player $i$. It can be computed more efficiently as $\max_s Q_i(s) - Q_i(s,s)$, where the marginal distribution $Q_{S,i}(s)$ is given by

$$Q_{S,i}(s) = \mathbb{P}(v(S) = s) = e^{\beta_s - \bar{\beta}_0}.$$

## B  EXTENDED DERIVATION FOR CATEGORICAL GAMES

We provide a derivation of the expressions $\tilde{Q}_{i,S}(r,s)$. In this derivation, $i$ and $S$ are fixed, and we write $\mathcal{P}_{rs}$ for $\tilde{Q}_{i,S}(r,s)$. Let $d \geq 3$ be an integer, $[\alpha_j]$ and $[\beta_j]$ be sets of $d$ real numbers. Above, $\alpha_j = \theta_{S \cup i,j}$ and $\beta_j = \theta_{S,j}$, but the derivation below does not make use of this. Also, let $\varepsilon_j$ be $d$ independent standard Gumbel variables, each of which has distribution function and density

$$F(\varepsilon) = \exp\left(e^{-\varepsilon}\right), \quad p(\varepsilon) = F(\varepsilon)' = \exp\left(-\varepsilon - e^{-\varepsilon}\right) = e^{-\varepsilon}F(\varepsilon).$$

Fix $r, s \in \{1, \ldots, d\}$, $r \neq s$. We would like to obtain an expression for the probability $\mathcal{P}_{rs}$ of

$$\arg\max_j (\alpha_j + \varepsilon_j) = r \quad \text{and} \quad \arg\max_j (\beta_j + \varepsilon_j) = s.$$

Define

$$\alpha_{jr} := \alpha_j - \alpha_r, \quad \beta_{js} := \beta_j - \beta_s.$$

The $\arg\max$ equalities above can also be written as a set of $2d$ inequalities (2 of which are trivial):

$$\varepsilon_j \leq \varepsilon_r - \alpha_{jr}, \quad \varepsilon_j \leq \varepsilon_s - \beta_{js}, \quad j = 1, \ldots, d.$$

Then:

$$\mathcal{P}_{rs} = \mathbb{E}\left[\prod_j I_j\right], \quad I_j := \mathbf{1}_{\varepsilon_j \leq \min(\varepsilon_r - \alpha_{jr}, \varepsilon_s - \beta_{js})}.$$

Two of them are simple:

$$I_r = \mathbf{1}_{\varepsilon_r \leq \varepsilon_s - \beta_{rs}}, \quad I_s = \mathbf{1}_{\varepsilon_s \leq \varepsilon_r - \alpha_{sr}}, \quad I_r I_s = \mathbf{1}_{\alpha_s - \alpha_r \leq \varepsilon_r - \varepsilon_s \leq \beta_s - \beta_r}.$$

Denote

$$\gamma_j := \alpha_{jr} - \beta_{js} = \rho_j - (\alpha_r - \beta_s), \quad \rho_j := \alpha_j - \beta_j.$$

Note that $\gamma_j$ depends on $r, s$, but $\rho_j$ does not. If $j \neq r, s$, then

$$I_j = \mathbf{1}_{\varepsilon_j \leq \varepsilon_r - \alpha_{jr}}\mathbf{1}_{\varepsilon_r - \varepsilon_s \leq \gamma_j} + \mathbf{1}_{\varepsilon_j \leq \varepsilon_s - \beta_{js}}\mathbf{1}_{\varepsilon_r - \varepsilon_s \geq \gamma_j}.$$

If we exchange sum and product, we obtain an expression of $\mathcal{P}_{rs}$ as sum of $2^{d-2}$ terms. Each of these terms is an expectation over $\varepsilon_r$, $\varepsilon_s$, with the argument being the product of $d-2$ terms $F(\varepsilon_r + a_j)$ or $F(\varepsilon_s + a_j)$ and a box indicator for $\varepsilon_r - \varepsilon_s$. In the sequel, we make this more concrete and show that at most $d-1$ of these terms are nonzero.

With a bit of hindsight, we assume that $\rho_1 \geq \rho_2 \geq \cdots \geq \rho_d$, which is obtained by reordering the categories. This implies that $[\gamma_j]$ is nonincreasing for all $(r, s)$. Also, define the function $\pi(k) = k + \mathbf{1}_{r \leq k} + \mathbf{1}_{s-1 \leq k}$ from $\{1, \ldots, d-2\}$ to $\{1, \ldots, d\} \setminus \{r, s\}$. We will argue in terms of a recursive computation over $k = 1, \ldots, d-2$. Define

$$M_k(\varepsilon_r, \varepsilon_s) = \mathbb{E}\left[ I_r I_s \prod\nolimits_{1 \leq j \leq k} I_{\pi(j)} \mid \varepsilon_r, \varepsilon_s \right], \quad k \geq 0,$$

so that $\mathcal{P}_{rs} = \mathbb{E}[M_{d-2}(\varepsilon_r, \varepsilon_s)]$. Each $M_k$ can be written as sum of $2^k$ terms. Imagine a binary tree of depth $d-1$, with layers indexed by $k = 0, 1, \ldots, d-2$. Each node in this tree is annotated by a box indicator for $\varepsilon_r - \varepsilon_s$ and some information detailed below. We are interested in the $2^{d-2}$ leaf nodes of this tree.

## B.1 Box Indicators. Which Terms are Needed?

We begin with a recursive computation of the box indicators, noting that we can eliminate all nodes where the box is empty. Label the root node (at $k = 0$) by 1, its children (at $k = 1$) by 10 (left), 11 (right), and so on, and define the box indicators as $\mathbf{1}_{l_1 \leq \varepsilon_r - \varepsilon_s \leq u_1}$, and $(l_{10}, u_{10})$, $(l_{11}, u_{11})$ respectively. Then, $l_1 = \alpha_s - \alpha_r$, $u_1 = \beta_s - \beta_r$ defines the box for the root. Here,

$$l_1 \geq u_1 \quad \Leftrightarrow \quad \rho_s \geq \rho_r.$$

Since $[\rho_j]$ is non-increasing, the root box is empty if $s < r$, so that $\mathcal{P}_{rs} = 0$ in this case. In the sequel, we assume that $r < s$ and $\rho_r > \rho_s$, so that $l_1 < u_1$.

If $\mathbf{n}$ is the label of a node at level $k-1$ with box $(l_{\mathbf{n}}, u_{\mathbf{n}})$, then

$$l_{\mathbf{n}0} = l_{\mathbf{n}}, \quad u_{\mathbf{n}0} = \min(\gamma_{\pi(k)}, u_{\mathbf{n}}), \quad l_{\mathbf{n}1} = \max(\gamma_{\pi(k)}, l_{\mathbf{n}}), \quad u_{\mathbf{n}1} = u_{\mathbf{n}}.$$

Consider node 11 (right child of root). There are two cases. (1) $\gamma_{\pi(1)} < u_1$. Then, $l_{11} \geq \gamma_{\pi(1)} \geq \gamma_{\pi(k)}$ for all $k \geq 1$, so all descendants must have the same $l = l_{11}$. If ever we step to the left from here, $u = \min(\gamma_{\pi(k)}, u_1) \leq \gamma_{\pi(k)} \leq \gamma_{\pi(1)} \leq l_{11}$, so the node is eliminated. This means from 11, we only step to the right: $111, 1111, \ldots$, with $l = \max(\gamma_{\pi(1)}, l_1)$, $u = u_1$, so there is only one leaf node which is a descendant of 11. (2) $\gamma_{\pi(1)} \geq u_1$. Then, $l_{11} \geq u_{11}$, so that 11 and all its descendants are eliminated.

At node 10, we have $l_{10} = l_1$. If $\gamma_{\pi(1)} \leq l_1$, the node is eliminated, so assume $\gamma_{\pi(1)} > l_1$, and $u_{10} = \min(\gamma_{\pi(1)}, u_1)$. Consider its right child 101. We can repeat the argument above. There is at most one leaf node below 101, with $l = \max(\gamma_{\pi(2)}, l_1)$ and $u = u_{10} = \min(\gamma_{\pi(1)}, u_1)$.

All in all, at most $d - 1$ leaf nodes are not eliminated, namely those with labels $10 \ldots 01 \ldots 1$, and their boxes are $[\max(\gamma_{\pi(1)}, l_1), u_1]$, $[\max(\gamma_{\pi(2)}, l_1), \min(\gamma_{\pi(1)}, u_1)]$, $\ldots$, $[\max(\gamma_{\pi(d-2)}, l_1), \min(\gamma_{\pi(d-3)}, u_1)]$, $[l_1, \min(\gamma_{\pi(d-2)}, u_1)]$.

Recall that each node term is a product of $d-2$ Gumbel CDFs times a box indicator. What are these products for our $d-1$ non-eliminated leaf nodes? The first is $F(\varepsilon_s - \beta_{\pi(1)s}) \cdots F(\varepsilon_s - \beta_{\pi(d-2)s})$, the second is $F(\varepsilon_r - \alpha_{\pi(1)r}) F(\varepsilon_s - \beta_{\pi(2)s}) \cdots F(\varepsilon_s - \beta_{\pi(d-2)s})$, the third is $F(\varepsilon_r - \alpha_{\pi(1)r}) F(\varepsilon_r - \alpha_{\pi(2)r}) F(\varepsilon_s - \beta_{\pi(3)s}) \cdots F(\varepsilon_s - \beta_{\pi(d-2)s})$ and the last one is $F(\varepsilon_r - \alpha_{\pi(1)r}) \cdots F(\varepsilon_r - \alpha_{\pi(d-2)r})$. Next, we derive expressions for the expectation of these terms.

## B.2 Analytical Expressions for Expectations

Consider $d-2$ scalars $a_1, \ldots, a_{d-2}$ and $1 \leq k \leq d-1$. We would like to compute

$$A = \mathbb{E}\left[ \left( \prod\nolimits_{j < k} F(\varepsilon_r + a_j) \right) \left( \prod\nolimits_{j \geq k} F(\varepsilon_s + a_j) \right) \mathbf{1}_{l \leq \varepsilon_r - \varepsilon_s \leq u} \right]. \tag{4}$$

Denote

$$G(a_1, \ldots, a_t) := \mathbb{E}[F(\varepsilon_1 + a_1) \cdots F(\varepsilon_1 + a_t)].$$

We start with showing that

$$G(a_1, \ldots, a_t) = \left(1 + e^{-a_1} + \cdots + e^{-a_t}\right)^{-1}.$$

Recall that $p(x) = F(x)' = e^{-x}F(x)$. If $\tilde{F}(x) = \prod_{j=1}^{t} F(x + a_j)$, then

$$\tilde{F}(x)' = \left(\sum_{j=1}^{t} e^{-a_j}\right) e^{-x}\tilde{F}(x).$$

Using integration by parts:

$$G(a_1, \ldots, a_t) = \int \tilde{F}(x)p(x)\, dx = 1 - \int \tilde{F}(x)'F(x)\, dx = 1 - \left(\sum_{j=1}^{t} e^{-a_j}\right) G(a_1, \ldots, a_t),$$

where we used that $F(x) = e^x p(x)$.

Next, define

$$g_1 = \log\left(1 + e^{-a_1} + \cdots + e^{-a_{k-1}}\right), \quad g_2 = \log\left(1 + e^{-a_k} + \cdots + e^{-a_{d-2}}\right).$$

We show that $A$ in (4) can be written in terms of $(g_1, g_2, l, u)$ only. Assume that $k > 1$ for now. Fix $\varepsilon_s$ and do the expectation over $\varepsilon_r$. Note that $\mathbf{1}_{l \leq \varepsilon_r - \varepsilon_s \leq u} = \mathbf{1}_{\varepsilon_s + l \leq \varepsilon_r \leq \varepsilon_s + u}$. If $\tilde{F}(x) = \prod_{j<k} F(x + a_j)$, then

$$\tilde{F}(x)' = \left(\sum_{j<k} e^{-a_j}\right) e^{-x}\tilde{F}(x).$$

Using integration by parts:

$$B(\varepsilon_s) = \int_{\varepsilon_s + l}^{\varepsilon_s + u} \tilde{F}(x)p(x)\, dx = \left[\tilde{F}(x)F(x)\right]_{\varepsilon_s + l}^{\varepsilon_s + u} - B(\varepsilon_s)\sum_{j<k} e^{-a_j},$$

so that

$$B(\varepsilon_s) = e^{-g_1}\left[\tilde{F}(x)F(x)\right]_{\varepsilon_s + l}^{\varepsilon_s + u}$$

and

$$A = \mathbb{E}\left[B(\varepsilon_s)\prod_{j \geq k} F(\varepsilon_s + a_j)\right] = A_1 - A_2,$$

where

$$A_1 = e^{-g_1}\mathbb{E}\left[\left(\prod_{j<k} F(\varepsilon_s + u + a_j)\right)\left(\prod_{j \geq k} F(\varepsilon_s + a_j)\right) F(\varepsilon_s + u)\right]$$
$$= e^{-g_1}G(a_1 + u, a_2 + u, \ldots, a_{k-1} + u, a_k, \ldots, a_{d-2}, u)$$

and

$$A_2 = e^{-g_1}G(a_1 + l, a_2 + l, \ldots, a_{k-1} + l, a_k, \ldots, a_{d-2}, l).$$

Now,

$$-\log A_1 = g_1 - \log G(a_1 + u, a_2 + u, \ldots, a_{k-1} + u, a_k, \ldots, a_{d-2}, u)$$
$$= g_1 + \log\left(1 + \sum_{j<k} e^{-a_j - u} + \sum_{j \geq k} e^{-a_j} + e^{-u}\right) = g_1 + \log\left(e^{g_2} + e^{-u+g_1}\right)$$
$$= g_1 + g_2 + \log\left(1 + e^{g_1 - g_2 - u}\right)$$

and

$$-\log A_2 = g_1 + g_2 + \log\left(1 + e^{g_1 - g_2 - l}\right)$$

so that

$$A = A_1 - A_2 = e^{-(g_1 + g_2)}\left(\sigma(g_2 - g_1 + u) - \sigma(g_2 - g_1 + l)\right), \quad \sigma(x) := \frac{1}{1 + e^{-x}}. \quad (5)$$

If $k = 1$, we can flip the roles of $\varepsilon_r$ and $\varepsilon_s$ by $g_1 \leftrightarrow g_2$, $l \to -u$, $u \to -l$, $k \to d - 1$, which gives

$$e^{-(g_1 + g_2)}\left(\sigma(-(g_2 - g_1 + l)) - \sigma(-(g_2 - g_1 + u))\right) = e^{-(g_1 + g_2)}\left(\sigma(g_2 - g_1 + u) - \sigma(g_2 - g_1 + l)\right),$$

using $\sigma(-x) = 1 - \sigma(x)$, so the expression holds in this case as well.

### B.3 Efficient Computation for All Pairs

Our $d-1$ terms of interest can be indexed by $k = 1, \ldots, d-1$. We can use the analytical expression just given with $a_j = -\alpha_{\pi(j)r}$ for $1 \leq j < k$ and $a_j = -\beta_{\pi(j)s}$ for $k \leq j \leq d-2$. Define

$$g_1(k) = \log\left(1 + \sum\nolimits_{1 \leq j < k} e^{\alpha_{\pi(j)} - \alpha_r}\right), \quad g_2(k) = \log\left(1 + \sum\nolimits_{k \leq j \leq d-2} e^{\beta_{\pi(j)} - \beta_s}\right),$$

as well as

$$l(k) = \max(\gamma_{\pi(k)}, l_1), \quad u(k) = \min(\gamma_{\pi(k-1)}, u_1),$$

where we define $\pi(0) = 0$, $\pi(d-1) = d+1$, $\gamma_0 = +\infty$, and $\gamma_{d+1} = -\infty$. Note that

$$\begin{aligned}
l(k) &= \max(\rho_{\pi(k)} - \alpha_r + \beta_s, \alpha_s - \alpha_r) = \beta_s - \alpha_r + \max(\rho_{\pi(k)}, \rho_s), \\
u(k) &= \min(\rho_{\pi(k-1)} - \alpha_r + \beta_s, \beta_s - \beta_r) = \beta_s - \alpha_r + \min(\rho_{\pi(k-1)}, \rho_r).
\end{aligned} \tag{6}$$

$\mathcal{P}_{rs}$ is obtained as sum of $A(g_1(k), g_2(k), l(k), u(k))$ for $k = 1, \ldots, d-1$. In the sequel, we show how to compute these terms efficiently, for all pairs $r < s$.

Recall that $\gamma_j = \rho_j - (\alpha_r - \beta_s)$, $u_1 = \beta_s - \beta_r$, $l_1 = \alpha_s - \alpha_r$. Then:

$$l(k) < u(k) \quad \Leftrightarrow \quad \rho_{\pi(k)} < \rho_{\pi(k-1)} \wedge \rho_{\pi(k)} < \rho_r \wedge \rho_s < \rho_{\pi(k-1)}.$$

Recall that $\pi(k) = k + \mathbf{1}_{r \leq k} + \mathbf{1}_{s-1 \leq k}$. Define $K_1 = \{1, \ldots, r-1\}$, $K_3 = \{s, \ldots, d-1\}$, each of which can be empty. For $k \in K_1$, $\rho_{\pi(k)} = \rho_k \geq \rho_r$, so $l(k) \geq u(k)$. For $k \in K_3$, we have $\pi(k-1) = k+1 > s$, so that $\rho_s \geq \rho_{\pi(k-1)}$ and $l(k) \geq u(k)$. This means we only need to iterate over $k \in K_2 = \{r, \ldots, s-2\}$ with $\pi(k) = k+1$ and $k = s-1$ with $\pi(k) = s+1$ (the latter only if $s < d$).

As $k$ runs in $K_2$, $\pi(k) = r+1, \ldots, s-1$, and if $s < d$ then $\pi(s-1) = s+1$. Now

$$g_1(k) = \log\left(1 + \sum\nolimits_{1 \leq j < k} e^{\alpha_{\pi(j)} - \alpha_r}\right) = \log\sum\nolimits_{1 \leq j \leq k} e^{\alpha_j - \alpha_r},$$

using that $e^{\alpha_r - \alpha_r} = 1$. For $g_2(k)$, if $k < s-1$, then $\{\pi(j) \mid k \leq j \leq d-2\} = \{k+1, \ldots, d\} \setminus \{s\}$, and if $k = s-1$, the same holds true (the set is empty if $s = d$). Using $e^{\beta_s - \beta_s} = 1$, we have

$$g_2(k) = \log\sum\nolimits_{k < j \leq d} e^{\beta_j - \beta_s}.$$

Define

$$\bar{\alpha}_k := \log\sum_{j=1}^{k} e^{\alpha_j}, \quad \bar{\beta}_k := \log\sum_{j=k+1}^{d} e^{\beta_j}, \quad k = 1, \ldots, d-1.$$

Then:

$$g_1(k) = \bar{\alpha}_k - \alpha_r, \quad g_2(k) = \bar{\beta}_k - \beta_s, \quad k = r, \ldots, s-1.$$

Finally, using $g_2(k) - g_1(k) = \bar{\beta}_k - \bar{\alpha}_k + \alpha_r - \beta_s$ and (6), we have

$$g_2(k) - g_1(k) + l(k) = \bar{\beta}_k - \bar{\alpha}_k + \max(\rho_{\pi(k)}, \rho_s), , \quad g_2(k) - g_1(k) + u(k) = \bar{\beta}_k - \bar{\alpha}_k + \min(\rho_{\pi(k-1)}, \rho_r).$$

Some extra derivation, distinguishing between (a) $r = s-1$, (b) $r < s-1 \wedge k \in K_2$, (c) $r < s-1 \wedge k = s-1$ shows that

$$\max(\rho_{\pi(k)}, \rho_s) = \rho_{k+1}, \quad \min(\rho_{\pi(k-1)}, \rho_r) = \rho_k, \quad k = r, \ldots, s-1.$$

Plugging this into (5):

$$A(k) = e^{\alpha_r + \beta_s} c_k, \quad c_k = e^{-\bar{\beta}_k - \bar{\alpha}_k}\left(\sigma\left(\bar{\beta}_k - \bar{\alpha}_k + \rho_k\right) - \sigma\left(\bar{\beta}_k - \bar{\alpha}_k + \rho_{k+1}\right)\right).$$

and $\mathcal{P}_{rs} = \sum_{k=r}^{s-1} A(k)$. Importantly, $c_k$ does not depend on $r, s$. Therefore:

$$\mathcal{P}_{rs} = e^{\alpha_r + \beta_s}(C_s - C_r), \quad C_t = \sum_{k=1}^{t-1} c_k \quad (r < s); \quad \mathcal{P}_{rs} = 0 \quad (r > s). \tag{7}$$

The sequences $[\bar{\alpha}_k], [\bar{\beta}_k], [c_k], [C_k]$ can be computed in $\mathcal{O}(d)$.

Finally, we also determine $\mathcal{P}_{rr}$, which is defined by the inequalities $\varepsilon_j \leq \varepsilon_1 - \max(\alpha_{jr}, \beta_{jr})$. A derivation like above (but simpler) gives:

$$\mathcal{P}_{rr} = \left(1 + \sum_{j \neq r} e^{\max(\alpha_{jr}, \beta_{jr})}\right)^{-1}.$$

Now, $\alpha_{jr} \geq \beta_{jr}$ iff $\rho_j \geq \rho_r$ iff $j < r$, so that

$$\mathcal{P}_{rr} = \left(1 + \sum_{j<r} e^{\alpha_j - \alpha_r} + \sum_{j>r} e^{\beta_j - \beta_r}\right)^{-1} = \left(e^{\bar{\alpha}_r - \alpha_r} + e^{\bar{\beta}_r - \beta_r}\right)^{-1} = e^{\beta_r - \bar{\beta}_r}\sigma(\bar{\beta}_r - \bar{\alpha}_r + \rho_r), \quad (r < d),$$

$$\mathcal{P}_{dd} = e^{\alpha_d - \bar{\alpha}_d}.$$

## C  RELATED WORK IN COOPERATIVE GAME THEORY.

The Shapley value of simple game has a probabilistic interpretation (Peleg & Sudhölter, 2007, pag. 168) however simple games are not Categorical games. An and-or axiom substitute the linear axioms in simple games (Weber, 1988), here we address probabilisitc combinations. Stochastic games are typically intended as multi-stage games where the transition between stages is stochastic Shapley (1953b); Petrosjan (2006) and not the intrinsic payoffs. Static cooperative games with stochastic output have been considered from the perspective of coalition formation and considering notions of players' utility (e.g. Suijs et al., 1999) or studying two stages setups – before and after the realisation of the payoff (e.g. Granot, 1977), and from an optimization perspective (Sun et al., 2022). To the best of our knowledge, our settings and constructions have not been studied before.

