# OpenReview forum: "Explaining Multiclass Classifiers with Categorical Values: A Case Study in Radiography"
_ICLR.cc/2023/Workshop/TML4H — ICLR 2023 Workshop TML4H Oral_

### Official Review · Reviewer_LrSE · 2023-03-01
**Interesting work, need more experimental results.**

**Rating:** 7
**Confidence:** 3

**Review:**

This paper proposes categorical shapley value to bridge the corelation between the feature and final output. The proposed method is applied to the classification of the pneumonia detection and subtyping from chest X-ray images.
The authors present three examples to demonstrate the effectiveness of the proposed method.

Overall, this paper is well-written and the proposed method is convincing. I have some minor comments as follows:
1. Quantitative evaluation is not provided. It would be better to provide quantitative evaluation to demonstrate the effectiveness of the proposed method.
2. The proposed method is applied to the classification of the pneumonia detection and subtyping from chest X-ray images. It would be better to provide more details about the dataset and the experimental setting.
3. The authors demonstrate the proposed method can be applied to versatile tasks. It would be better to provide more examples to demonstrate the effectiveness of the proposed method.

---

### Official Review · Reviewer_LSsr · 2023-03-03
**This paper is well-structured with significant contributions.**

**Rating:** 8
**Confidence:** 3

**Review:**

In this paper, the authors proposed "Categorical Shapley value" as a theoretically grounded method to explain the output of multi-class classifiers. Additionally, a X-ray image case was studied to demonstrate  its the explainability.

Pros:
1. The theoretical parts on P2 and P3 are well organized, with clear reference supports;
2. The definition of  categorical games and  values are clear.

Cons:
1. Some sentences in this paper is not very accurate, for instance, on p1 "However,
the adoption of ML in clinical practice has often been hampered by the opaqueness of the ML
models." While there are some explainable ML models.
2. The case study’s explanation can be more sufficient.

---

### Meta-Review · Area_Chair_RkuN · 2023-03-05

**Recommendation:** Accept (Oral)
**Confidence:** 4

**Metareview:**

This paper is well structured and tackles important and commonly used Shapley values.
There is clear added value and should be relevant to readers for the conference.